# Scaling laws in Hall-inertial range turbulence

Yasuhito Narita[1], Wolfgang Baumjohann[1], and Rudolf A. Treumann[2,3]

[1]Space Research Institute, Austrian Academy of Sciences, Schmiedlstraße 6, A-8042 Graz, Austria
[2]International Space Science Institute, Hallerstraße 6, CH-3012, Berne, Switzerland
[3]Geophysics Department, Ludwig-Maximilians-Universität, Theresienstraße 41, D-80333 Munich, Germany

**Correspondence:** Y. Narita
(yasuhito.narita@oeaw.ac.at)

**Abstract.** There is an increasing amount of observational evidence in space plasmas for the breakdown of inertial-range spectra of magnetohydrodynamic (MHD) turbulence on spatial scales smaller than the ion inertial length. Magnetic energy spectra often exhibit a steepening, which is reminiscent of dissipation of turbulence energy, for example in wave-particle interactions. Electric energy spectra, on the other hand, tend to be flatter than those of MHD turbulence, which is indicative of a dispersive process converting magnetic into electric energy in electromagnetic wave excitation. Here we develop a model of the scaling laws and the power spectra for the Hall-inertial range in plasma turbulence. In the present paper we consider a two-dimensional geometry with no wave vector component parallel to the magnetic field as is appropriate in Hall-MHD. A phenomenological approach is taken. The Hall electric field attains an electrostatic component when the wave vectors are perpendicular to the mean magnetic field. The power spectra of Hall-turbulence are steep for the magnetic field with slope of $-7/3$ for compressible magnetic turbulence, they are flatter for the Hall electric field with slope $-1/3$. Our model for the Hall-turbulence gives a possible explanation for the steepening of the magnetic energy spectra in the solar wind neither as indication of the dissipation range nor the dispersive range but as the Hall-inertial range. Our model also reproduces the shape of energy spectra in Kelvin-Helmholtz turbulence observed at the Earth magnetopause.

## 1 Introduction

The recent availability of multi-spacecraft missions such as Cluster (Escoubet et al., 2001), THEMIS (Angelopoulos, 2008), and MMS (Burch et al., 2016) together with substantial advances in their instrumentation and the subsequent data analysis opened up the door to a more detailed study of space plasma turbulence on ion-scales when ion-inertia comes into play. On those scales ions demagnetise and ultimately decouple magnetically from the electron motion. The drop-out of ions from magnetic dynamics necessarily leaves its signature in the turbulent power spectra, possibly causing deviations from the conventionally accepted inertial range slopes of turbulence. Typical ion-inertial range scale lengths in the solar wind range from 100 km to 1000 km, which correspond to the turbulent wavenumber interval $10^{-6} \lesssim (2\pi)^{-1} k \lesssim 10^{-5}$ m$^{-1}$. Multiplying, for instance, with a nominal solar wind speed of $\sim 500$ km s$^{-1}$ this interval maps to the frequency range $0.5 \lesssim (2\pi)^{-1} \omega \lesssim 5$ Hz.

Physically speaking, in a medium of density $n$ moving at velocity $\boldsymbol{V}$ the main signature of the ion-inertial range, sometimes also called the ion-kinetic regime or ion-dissipation range, is the presence of Hall currents. These are pure electron currents $\boldsymbol{j}_H = -en\,\boldsymbol{E} \times \boldsymbol{B}/B^2$ flowing perpendicular to the magnetic $\boldsymbol{B}$ and convection electric $\boldsymbol{E} = -\boldsymbol{V} \times \boldsymbol{B}$ fields.

This ion-scale Hall-turbulence is, for example in the solar wind, two-dimensional with both wave vectors and fluctuating magnetic fields confined to the plane perpendicular with respect to the mean field $\boldsymbol{B}_0$. Ion-kinetic scale power-law spectra (though limited to the frequency domain) were observationally obtained separately for the magnetic (Alexandrova et al., 2009; Sahraoui et al., 2009) and electric (Bale et al., 2005) fields. They were found to be reminiscent of a turbulent inertial range typical for scale-invariant turbulence spectra of Richardson-Kolmogorov or Iroshnikov-Kraichnan type.

Dispersion analyses performed on the fluctuations showed the absence of any clear spectral eigenmodes (in linear Vlasov theory) which would result from dispersion relations in the presumable ion-scale wavenumber-frequency domain. At the best, there were rather weak indications found only of otherwise expected kinetic Alfvén, whistler, and ion Bernstein modes (Perschke et al., 2015, 2016; Roberts et al., 2015) in this range. The breakdown of linear-mode theory thus indicates that the frequencies deviate from simple Doppler-shifted linear modes by random sweeping which would be due to the large-scale variations of the flow such as eddies or Alfvénic fluctuations, sideband formation caused in both, a weakly turbulent kinetic wave-wave coupling, or steepening in the course of nonlinear evolution. Also solitary-structure formation resulting from phase coherence (Narita, 2018) seems to be absent. The study by Roberts et al. (2018) indicates the existence of the kinetic Alfvén mode in the magnetosheath region as obtained from the wave analysis for the fluctuations in the MMS data using the Alfvén ratio. No dispersion analysis is performed. On the other hand, the study by Narita et al. (2016) exhibits a frequency scattering in the observationally-determined dispersion relation with an indication of a kinetic-drift mirror mode.

Based on these observations, we consider in the following a phenomenological turbulence model of stationary inertial-range spectra evolving in ion-scale turbulence. We will show that, qualitatively, such a model reproduces ion-inertial range spectra measured by the MMS spacecraft in the vicinity of the magnetopause (Stawarz et al., 2016), whenever the Kelvin-Helmholtz-instability is excited and decays into smaller scale Kelvin-Helmholtz vortices until a spectrum of low frequency small-scale turbulence is produced. We consider a two-dimensional geometry which has no parallel wave vector component. The full expression for the Hall electric field contains also parallel wave vector components (Treumann et al., 2019) which in Hall MHD are neglected.

Limitations of Hall MHD have been discussed, for example, by Howes (2009). The concept of Hall turbulence is valid in the limit where the electron temperature is much greater than the ions temperature and when the inverse of the linear transit time for an ion is much smaller than the turbulent frequency and the inverse of the linear transit time for an electron, respectively. Thus, in the instance where the temperature of the ions is finite, phase-mixing and damping of modes ought to be taken into account. This causes deviations from Hall MHD.

The results of our endeavour can be summarised as follows: The Hall electric field attains the electrostatic component when the wave vectors are perpendicular to the mean magnetic field. Scaling laws are derived for the magnetic field and electric field in a power-law form. In the case of the compressible magnetic field fluctuations (with the parallel fluctuations of the magnetic field), the energy spectra have a slope of $-7/3$ and $-1/3$ for the magnetic field and the Hall electric field, respectively. The

amplitude ratio of the Hall electric field to the fluctuating magnetic field (hereafter, the E-B ratio) has a linear dependence on the wavenumber. The density power spectrum has a positive spectral slope with an index of $+5/3$. In the incompressible case with the perpendicular fluctuations of the magnetic field, the energy spectra have a slope of $-2$ and $-1$ for the magnetic and electric fields, respectively. The E-B ratio has a dependence of the wavenumber with a power of $1/2$. An important lesson from the model construction is that the Hall electric field is dependent on the wavenumber and the E-B ratio also shows the wavenumber dependence.

## 2 The Hall fluctuation fields

### 2.1 The Hall electric fields

Separation of ion and electron motion in a streaming magnetised plasma generates a Hall current $\boldsymbol{j}_H$. Referring to the magnetised electron equation of motion, which is the generalised collisionless Ohm's law for the electric field, the Hall current produces its specific Hall-electric field

$$\boldsymbol{E}_H = -\frac{1}{en}\boldsymbol{j}_H \times \boldsymbol{B} \tag{1}$$

The Hall current $\boldsymbol{j}_H$ is perpendicular to the magnetic and electric fields. In turbulence theory one is interested in the fluctuations of the fields given in the form $\delta \boldsymbol{F} = \boldsymbol{F} - \boldsymbol{F}_0$, with $\boldsymbol{F}$ referring to any relevant turbulent field, magnetic, electric, flow velocity, density, and so on, denoting fluctuations with prefix $\delta$ and, in the following suppressing the index $0$ on all mean fields. Since the electrons remain magnetic and continue their mean flow, the mean Hall electric field is just the mean convection electric field of the flow $\boldsymbol{E}_{H0} = \boldsymbol{E}_0 = -\boldsymbol{V}_0 \times \boldsymbol{B}_0$ and, hence, the mean Hall current is $\boldsymbol{j}_{H0} = -en_0\boldsymbol{E}_{H0}$, which is of no interest. Due to their inertia the ions continue to participate in the flow also on those scales in the collisionless plasma. Thus the zero-th order Hall current can be neglected when considering fluctuations in turbulence. The fluctuation of the Hall current taken in the moving frame $\boldsymbol{V}_0 = \boldsymbol{E}_0 = 0$ is obtained as

$$\delta \boldsymbol{j}_H = -e\delta\left[n\frac{\boldsymbol{E} \times \boldsymbol{B}}{B^2}\right]_H \simeq -en\frac{\delta \boldsymbol{E}_H}{B} \times \frac{\boldsymbol{B}}{B}. \tag{2}$$

Electric field variations contribute primarily to the linear Hall current fluctuations, with fluctuating density and magnetic field contributions being of higher order. As expected the turbulent Hall current lies in the plane perpendicular to the mean field $\boldsymbol{B}$ and is perpendicular to the fluctuation in the Hall electric field. In the stationary observer's frame there would be a number of other terms which, however, disappear in the moving frame, the case which we are interested in here. There would also be higher-order Hall current terms when folding with the first-order magnetic and electric field fluctuations which we neglect to lowest order here.

### 2.2 Relations between the electric and magnetic fields

The magnetic field fluctuations $\delta \boldsymbol{B}_H$ in Hall turbulence have two components, one compressive component $\delta B_\parallel$ parallel to the mean field $\boldsymbol{B}$ and the other perpendicular component $\delta B_\perp$. It is convenient to introduce an orthogonal coordinate system

with base vectors $e_1$ and $e_2$ perpendicular to the mean field, and $e_\parallel$ along the mean field. Moreover, we are free to choose the direction of the perpendicular wave vector, letting $e_1$ refer to $k_\perp$. The Hall magnetic field has no divergence, so it must be perpendicular to $k$. This yields

$$\delta \boldsymbol{B} = (0, \delta B_\perp, \delta B_\parallel).$$

The fluctuation of the Hall electric field is given by

$$\delta \boldsymbol{E}_H = \frac{1}{en} \delta \boldsymbol{j}_H \times \boldsymbol{B} - \frac{\delta n}{n} \boldsymbol{E}. \tag{3}$$

The last term on the right containing the fluctuations in density and their contribution to $\delta \boldsymbol{E}_H$ is important only in the stationary frame where $\boldsymbol{E} \neq 0$. Using Ampère's law $\mu_0 \delta \boldsymbol{j} = \nabla \times \delta \boldsymbol{B}$ (from here on suppressing the index $H$ on the fluctuations when dealing exclusively with Hall fluctuations in Hall MHD) yields

$$\delta \boldsymbol{E} = \frac{1}{en\mu_0} \boldsymbol{B} \times (\nabla \times \delta \boldsymbol{B}). \tag{4}$$

It follows from Eq. (4) that in both cases the Hall-electric field is along the perpendicular Hall wave vector, i.e., along $e_1$. This shows that the fluctuation part of the Hall electric field is purely electrostatic, a property of which we can make use below. Switching to the Fourier representation with $\nabla \to i\boldsymbol{k}$ we obtain

$$\delta E_\perp^{(c)} \quad = \quad \frac{\mathrm{i}}{en\mu_0} k_\perp \delta B_\parallel B \tag{5}$$

$$\delta E_\perp^{(i)} \quad = \quad \frac{i}{en\mu_0} k_\perp |\delta B_\perp|^2 \tag{6}$$

for the compressive $\delta E^{(c)}$ and incompressible $\delta E^{(i)}$ components of the wavenumber-parallel (i.e., longitudinal fluctuation sense) electric field fluctuations, respectively. This shows that the incompressible electric field is second order in the magnetic fluctuation and will in principle be small and negligible. The ratio of the two components

$$\frac{\delta E_\perp^{(i)}}{\delta E_\perp^{(c)}} = \frac{|\delta B_\perp|^2}{B \delta B_\parallel} \tag{7}$$

depends on the sign of the compressive component of the magnetic field $\delta B_\parallel$. Since its right-hand side only contains magnetic components, it can be used in spacecraft observations to estimate the ratio on the left. With this information the dominant turbulent Hall-electric field is given by the compressive magnetic Hall field component, and the phase speed of the compressive Hall fluctuations is found to be

$$\frac{\delta E_\perp^{(c)}}{\delta B_\parallel} = \frac{ik_\perp B}{en\mu_0} = iV_A \frac{k_\perp c}{\omega_i} \tag{8}$$

with $V_A$ the Alfvén speed and $\omega_i$ the ion plasma frequency. Squaring this, we obtain

$$\left( \frac{\delta E_\perp^{(c)}}{V_A \delta B_\parallel} \right)^2 = \frac{k_\perp^2 c^2}{\omega_i^2}. \tag{9}$$

The right-hand side of this expression is the rudiment of either a very low frequency ion wave or ion whistler dispersion relation which for ion waves can be written as $k_\perp^2 c^2/\omega_i^2 = \omega^2/\omega_i^2 - 1$ and which is reproduced for $\omega = 0$. Thus the compressive part of Hall turbulence can be understood as the zero-frequency fluctuations of transverse ion waves

$$\frac{k_\perp^2 \tilde{V}_A^2}{\Omega_i^2} = 1 - \frac{\omega^2}{\Omega_i^2}\frac{V_A^2}{U^2} \tag{10}$$

in the limit $\omega \to 0$, where $U = \delta E_\perp^{(c)}/\delta B_\parallel$ is the complex phase speed, and $\tilde{V}_A = V_A^2/U$ is a modified Alfvén speed, which shows that these waves are essentially ion whistlers or modified zero frequency Alfvén waves. Resolving for the ficticious frequency $\omega$ one obtains

$$\frac{\omega}{\Omega_i} = \pm\frac{U}{V_A}\sqrt{1 + \frac{k_\perp^2 \tilde{V}_A^2}{\Omega_i^2}} \sim \pm i\frac{k_\perp c}{\omega_i}\eta \tag{11}$$

as can be shown by using the above expressions for $U$. Here $\eta \approx 0$ is a very small number. Nevertheless it is seen that, in principle, these waves would have linear dispersion $\omega \sim k_\perp$ if attaining any however small frequency. In addition, they would be damped.

What concerns the incompressible part, so its phase speed becomes a function of the transverse magnetic Hall field $\delta B_\perp$. We should note that the ratio of electric-to-magnetic fields is widely used in spacecraft observations in various plasma domains as an estimator of the plasma convective motion and the phase speed of the electromagnetic wave (Matsuoka et al., 1991; Bale et al., 2005; Eastwood et al., 2009).

## 2.3 Hall current related density fluctuations

Ions are non-magnetic. So, since the Hall field is electrostatic and the wavenumber and electric field are aligned, the ions respond to the presence of an electric Hall field via Poisson's equation to generate an electric-fluctuation related density fluctuation

$$\frac{\delta n}{n} = \frac{i\epsilon_0}{en}k_\perp \delta E_\perp^{(c)} = k_\perp^2 \frac{V_A^2}{\omega_i^2}\frac{\delta B_\parallel}{B}. \tag{12}$$

Here, only the compressive field component contributes because of its linearity. It shows that the relative density fluctuations are completely determined by the compressive Hall-magnetic field fluctuations $\delta B_\parallel$. This is an important conclusion as it shows that the turbulent density spectrum caused by the Hall effect is proportional to the turbulent compressive magnetic Hall spectrum whose wavenumber dependence is raised by the power of $k_\perp^4$, an effect which should become observable in the density spectrum in the scale range where ion inertia becomes susceptible. There the Hall modification of the density spectrum adds to the non-Hall deformation of the density spectrum derived in our former publication (Treumann et al., 2019). We will briefly return to this item below after having constructed the power spectrum of the magnetic fluctuations in the Hall field case.

## 3  Ion-scale inertial-range spectra

In order to proceed quantitatively, we need to construct wavenumber scaling laws for the spectra of the various field fluctuations. Subsequently we intend to determine the ratio $\delta E/\delta B$ as well as the energy spectra in an attempt to obtain a scaled model of the ion-inertial scale field fluctuations as the necessary step to derive the turbulent inertial-range power spectra of the fields on ion-inertial scale lengths $1 \lesssim kc/\omega_i < kc/\omega_e$ (where $\omega_e$ denotes the electron plasma frequency).

To this end we turn to the application of a phenomenological turbulence model (Biskamp et al., 1996) in two-dimensional electron magnetohydrodynamics which is appropriate in our case. We have already made use of two-fluid plasma theory above when referring to the presence of the Hall effect in the generalised Ohm's law. Let us introduce the following scaling

$$\delta B \propto \ell^{\alpha_m} \propto k^{-\alpha_m} \tag{13}$$

for the turbulent magnetic field. We, moreover, normalize all relevant fields and scales to the Alfvén speed $V_A$, ion cyclotron frequency $\Omega_i$, and mean magnetic field $B$ as follows:

$$\delta B \quad \to \quad \delta\tilde{B} = \frac{\delta B}{B} \tag{14}$$

$$\delta V \quad \to \quad \tilde{v} = \frac{\delta V}{V_A} \tag{15}$$

$$\delta E \quad \to \quad \delta\tilde{E} = \frac{\delta E}{V_A B} \tag{16}$$

$$\delta n \quad \to \quad \delta\tilde{n} = \frac{\delta n}{n} \tag{17}$$

$$k \quad \to \quad \tilde{k} = \frac{k V_A}{\Omega_i}. \tag{18}$$

### 3.1  Compressible magnetic turbulence

In the two-dimensional compressible turbulence configuration, the electron flow velocity is confined to the plane perpendicular to the mean magnetic field, but the magnetic field fluctuation $B_\parallel$ is compressible. The effect of the Hall effect on the fluctuation spectrum implies that the magnetic field becomes increasingly extended and compressed like an elastic spring. The flow velocity is determined by the gradient of the stream function as $\tilde{v} = (\boldsymbol{e}_\parallel \times \tilde{\nabla})\delta\tilde{B}_\parallel$, with the parallel fluctuation component $\delta\tilde{B}_\parallel$ of the magnetic field playing the role of a stream function. The eddy interaction time in units of the ion gyro-period $\Omega_i^{-1}$ becomes

$$\tilde{\tau} \propto (\tilde{k}\tilde{v})^{-1} \propto \tilde{k}^{-2}\delta\tilde{B}_\parallel^{-1} \tag{19}$$

For the energy transfer rate, which is assumed to be constant over the entire turbulent inertial range, we have

$$\tilde{\epsilon} \propto \frac{|\delta\tilde{B}_\parallel|^2}{\tilde{\tau}} \propto |\delta\tilde{B}_\parallel|^3 \tilde{k}_\perp^2. \tag{20}$$

This leads to the magnetic field scaling

$$\delta\tilde{B}_\parallel \sim c_m \tilde{\epsilon}^{1/3} \tilde{k}_\perp^{-2/3}, \tag{21}$$

where a proper scaling coefficient $c_m$ has been introduced. With these expressions, the magnetic energy spectrum becomes

$$\mathcal{E}_{\mathrm{mag}} = \frac{|\delta\tilde{B}_\parallel|^2}{\Delta\tilde{k}_\perp} \sim c_m^2 \tilde{\epsilon}^{2/3} \tilde{k}_\perp^{-7/3}, \tag{22}$$

where the wavenumber interval is scaled to the wavenumber itself as $\Delta\tilde{k}_\perp \sim \tilde{k}_\perp$, i.e. assuming an equidistant grid on the logarithmic scale. This $k^{-7/3}$ scaling of the magnetic energy spectrum is intriguing in view of the same scaling which had

been obtained in numerical simulations for isotropic Hall magnetohydrodynamic (Hall MHD) turbulence (Hori and Miura, 2008), the exact case which underlies our endeavour.

The energy spectrum for the Hall electric field fluctuation follows from the relation $\delta\tilde{E} = \tilde{k}_\perp \delta\tilde{B}_\parallel$ together with Eq. (21) as

$$\mathcal{E}_{\mathrm{elec}} = \frac{\tilde{k}_\perp^2 |\delta\tilde{B}_\parallel|^2}{\Delta\tilde{k}_\perp} \sim c_m^2 \tilde{\epsilon}^{2/3} \tilde{k}_\perp^{-1/3}. \tag{23}$$

These two expressions can be used to calculate the ratio $\delta E^{(c)}/\delta B$ of the fluctuation amplitudes

$$\left| \frac{\delta\tilde{E^{(c)}}}{\delta\tilde{B}_\parallel} \right| = \sqrt{\frac{\mathcal{E}_{\mathrm{elec}}}{\mathcal{E}_{\mathrm{mag}}}} \sim \tilde{k}_\perp, \tag{24}$$

which scales as the first power of the normalised wavenumber, indicating that the normalised fluctuation phase speed $\tilde{U}$ referred to linear increases above with decreasing scale in the ion-inertial range. Thus, this dependence remains unchanged and is in fact confirmed by the model.

Since for the turbulent Hall-velocity fluctuations we have $\tilde{v} = \delta\tilde{E}$, the kinetic energy spectrum $\mathcal{E}_{\mathrm{kin}}$ scales like the electric

power spectrum

$$\mathcal{E}_{\mathrm{kin}} = \mathcal{E}_{\mathrm{elec}} \tag{25}$$

As expected, the electric fluctuation spectrum in the Hall effect maps the kinetic fluctuation spectrum.

Finally coming to the spectrum of density fluctuations, we invoke Eq. (12) which in its rescaled form reads

$$\delta\tilde{n} = i \left( \frac{V_A}{c} \right)^2 \tilde{k}_\perp \delta\tilde{E}^{(c)}. \tag{26}$$

It yields the turbulent Hall-density power spectrum as

$$\mathcal{E}_{\mathrm{dens}} = \frac{|\delta\tilde{n}|^2}{\Delta\tilde{k}_\perp} \sim \left( \frac{V_A}{c} \right)^4 c_m^2 \tilde{\epsilon}^{2/3} \tilde{k}_\perp^{5/3}. \tag{27}$$

Most interestingly, this spectrum is of an *inverse* Kolmogorov type. Because of the relation between the Hall fluctuations in density, electric and magnetic fields, one of course expects that the presence of the Hall effect in the ion inertial range affects the shape of the density power spectrum. This is indeed the case. In the ion-inertial scale range the Hall effect seems

to practically compensate for the general spectral Kolmogorov slope of the density power spectrum, causing it to flatten substantially. Scaling-wise speaking, this is quite a strong effect, the degree of whose signature in observed density power spectra does, however, depend on the various scaling constants in the spectral contributions. One may, however, speculate that

the notoriously frequently observed $\tilde{k}^{-1}$ slope in the density power spectra in the solar wind around the presumable ion-inertial scale range, for example in Šafránková et al. (2015), may result from the contribution of the Hall effect to the inertial-range spectrum of ion-inertial scale turbulence.

It is interesting to compare the density spectrum with $k_\perp^{5/3}$ for the Hall-scaling (Eq. 27) with the Kolmogorov-Poisson density spectrum with the $k_\perp^{1/3}$-scaling obtained earlier (Treumann et al., 2019, Eq. 24) for non-Hall turbulence. The ratio of the two expressions is

$$\frac{\mathcal{E}_{\text{dens}}^H}{\mathcal{E}_{\text{dens}}^K} \sim \left(\frac{V_A}{c}\right)^2 \frac{c_m^2}{c_K} k_\perp^{4/3}. \tag{28}$$

It still depends on the unknown constant of proportionality $c_m$ which must be determined otherwise. However, the deformation of the spectral scaling caused by the Hall turbulence is stronger than in the non-Hall case. Its contribution might thus become important, even though numerically its contribution to the density variation is smaller than that of the Kolmogorov-Poisson spectrum, because $V_A \ll c$. The difference in the spectral slopes of $k_\perp^{4/3}$, indicates that the Hall density spectrum becomes increasingly more effective at larger wave numbers.

The Hall magnetic energy spectrum is steeper than the Kolmogorov-type one with wave number ratio

$$\frac{\mathcal{E}_{\text{mag}}^H}{\mathcal{E}_{\text{mag}}^K} \sim k_\perp^{-2/3} \tag{29}$$

Finally, the ratio of the kinetic power spectra yields a flatter Hall kinetic energy spectrum than Kolmogorov:

$$\frac{\mathcal{E}_{\text{kin}}^H}{\mathcal{E}_{\text{kin}}^K} \sim k_\perp^{4/3}. \tag{30}$$

## 3.2 Incompressible magnetic turbulence

In this section we briefly turn to the incompressible Hall spectra. As we had already noted, they play a lesser role in Hall turbulence for the quadratic dependence on the incompressible Hall-magnetic field fluctuation component $\delta B_\perp$ which would enable us to neglect it completely. However, for the sake of completeness we provide the corresponding expressions here below.

The scaling law in incompressible magnetic field fluctuations is determined by the estimate of the flow velocity in the perpendicular plane for the $\boldsymbol{E} \times \boldsymbol{B}$ drift motion of the electron fluid, $\tilde{v}_\perp = \delta \tilde{E} = \tilde{k}_\perp |\delta \tilde{B}_\perp|^2$. The time scale for the interaction is

$$\tilde{\tau} \propto \frac{\tilde{\ell}_\perp}{\tilde{v}_\perp} \propto \tilde{k}_\perp^{-2} \delta \tilde{B}_\perp^{-2}. \tag{31}$$

The energy transfer rate, again assumed to be constant in the inertial range, is

$$\tilde{\epsilon} \propto \frac{|\delta \tilde{B}_\perp|^2}{\tilde{\tau}} \propto \tilde{k}_\perp^2 |\delta \tilde{B}_\perp|^4. \tag{32}$$

The scaling law for the magnetic field follows from Eq. (32) as

$$\delta \tilde{B}_\perp \sim c_m \tilde{\epsilon}^{1/4} \tilde{k}_\perp^{-1/2}, \tag{33}$$

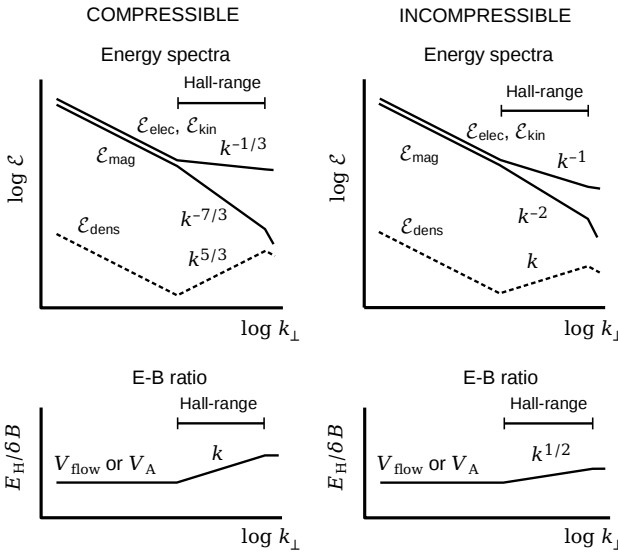

**Figure 1.** Top panels show the energy spectra for the magnetic field ($\mathcal{E}_{\mathrm{mag}}$), the Hall electric field ($\mathcal{E}_{\mathrm{elec}}$), and the flow velocity ($\mathcal{E}_{\mathrm{kin}}$), and the density fluctuation ($\mathcal{E}_{\mathrm{dens}}$) for compressible magnetic turbulence (top left panel) and incompressible turbulence (top right panel). Bottom panels are the E-B ratio for compressible magnetic turbulence (bottom left panel) and incompressible turbulence (bottom right panel).

and the magnetic energy spectrum reads

$$\tilde{\mathcal{E}}_{\mathrm{mag}} = \frac{|\delta\tilde{B}_{\perp}|^2}{\Delta\tilde{k}_{\perp}} \sim c_{\mathrm{m}}^2 \tilde{\epsilon}^{1/2} \tilde{k}_{\perp}^{-2}. \tag{34}$$

The energy spectrum for the Hall electric field and that for the kinetic energy have the same spectral forms because again $\tilde{u}_{\perp} = \delta\tilde{E}$, yielding

$$5 \quad \tilde{\mathcal{E}}_{\mathrm{elec}} = \tilde{\mathcal{E}}_{\mathrm{kin}} \sim c_m^4 \tilde{\epsilon} \tilde{k}_{\perp}^{-1}. \tag{35}$$

The ratio $\delta E^{(i)}/\delta B$ follows as

$$\left| \frac{\delta\tilde{E}^{(i)}}{\delta\tilde{B}_{\perp}} \right| \sim c_m \tilde{\epsilon}^{1/4} \tilde{k}_{\perp}^{1/2}. \tag{36}$$

Even though the magnetic fluctuation field is incompressible, the density varies because of the electrostatic nature of the Hall field. The density spectrum for the Hall electric field is

$$10 \quad \mathcal{E}_{\mathrm{dens}} \sim \left( \frac{V_A}{c} \right)^4 c_m \tilde{\epsilon} \tilde{k}_{\perp}. \tag{37}$$

Compared to the compressible case it increases at a lesser power with decreasing scale though also acting to compensate for the Kolmogorov slope.

Note that purely two-dimensional turbulence, in which the gradients, wave vectors, and field fluctuations are confined to the perpendicular plane, is rather improbable in electron-magnetohydrodynamics (EMHD) because the $\boldsymbol{E} \times \boldsymbol{B}$ drift motion causes the electric current in the perpendicular plane. It generates a parallel magnetic field fluctuation $\delta B_\parallel$ . This has been taken care of in the previous section which included the dominant Hall effect on the spectra.

Figure 1 shows the schematic shapes of the Hall field spectra in the ion inertial range for the two cases of compressible and incompressible Hall turbulence. Their absolute contribution to the observed turbulence depends on the proportionality factor $c_m$ which enters all above expressions. It is, however, seen that the kinetic energy in the Hall range dominates the magnetic energy. This is of course reasonable because the Hall effect is the result of the turbulence in the flow velocity. The dominant effect is provided by the compressive part of Hall turbulence. The compressive Hall magnetic spectrum decays slightly steeper

than the Kolmogorov spectrum. One therefore expects that observed magnetic spectra in stationary Hall turbulence in the ion-inertial-scale range will obey inertial range spectral indices $k^{-7/3}$. On the contrary, however, the Hall-density power spectra should exhibit a spectral increase in the ion-inertial scale Hall range which is due to the reaction of the density to the Hall electric turbulence. Naturally this effect is more strongly expressed in the compressible case.

Since the total turbulence spectra in the ion-inertial range are composed of the superposition of Hall and non-Hall contri-

butions it becomes fairly clear that the ion-inertial range spectra must deviate quite strongly from the inertial range spectra of hydrodynamic turbulence (Richardson-Kolmogorov) or that of hydromagnetic turbulence (Iroshnikov-Kraichnan).

## 4    Conclusion and discussions

The present communication dealt exclusively with the effect of the generation of Hall current turbulence in collisionless stationary homogeneous and isotropic inertial range magnetohydrodynamic turbulence on ion-inertial scales $1 \lesssim kc/\omega_i < kc/\omega_e$

where ion inertia takes over to determine the dynamics and ions de-magnetise. This magnetohydrodynamic range also refers to Hall-magnetohydrodynamics or electron magnetohydrodynamics. We first discussed in detail the appearance and properties of the Hall effect under conditions of interest in turbulence. We then switched and referred to a phenomenological scaling model. We derived the wavenumber scalings of the turbulent fluctuations and turbulent power spectral densities under Hall conditions. These investigations refer to the stationary frame of turbulence.

The first interesting result of this endeavour was that in stationary homogeneous turbulence the Hall contribution can be separated into compressive and non-compressive parts. It turned out that the compressive contribution to Hall turbulence dominates as it is first order in the turbulent magnetic field perturbation the Hall effect introduces. It was also found that the compressive Hall turbulence corresponds to kind of a zero-frequency ion-wave whose complex phase-speed is given by the ratio of electric and magnetic fluctuations. This phase speed increases with shrinking scale across the ion-inertial range being

linearly proportional to the turbulent wavenumber.

Knowing the relations between the turbulent field fluctuations under the conditions when the Hall effect has to be taken into account in collisionless stationary and homogeneous turbulence, we considered the turbulent inertial Hall state. Turning

to a dimensional analysis we were able to obtain the relevant scaling laws for the power spectral densities with respect to wavenumber holding in inertial-range Hall-turbulent power law spectra.

**Spectral shapes**

Transition to phenomenological electron-magnetohydrodynamics enabled the construction of the Hall-inertial range turbulent scaling laws on ion-inertia scales, an important and to our knowledge new finding which possibly enables the identification of the ion-inertial range from observation of magnetic, kinetic, and density turbulent power law shapes. For instance, the Hall-turbulence model qualitatively explains the Hall-range energy spectra of the Kelvin-Helmholtz-type turbulence at the magnetopause (Stawarz et al., 2016) in that (1) the (electron) flow velocity and the electric field exhibit the same spectral curve perpendicular to the mean magnetic field; and (2) the magnetic energy spectrum is markedly steeper than that of the kinetic energy and the electric field energy.

Compressive inertial-range Hall-magnetic power spectra scale like $\sim k_\perp^{-7/3}$, steeper than Richardson-Kolmogorov and Iroshnikov-Kraichnan while being, in some cases, in agreement with numerical simulations. This suggests that observed gradually increasing slopes in turbulent magnetic power spectra and becoming steeper at shorter scales than Kolmogorov may indicate that Hall turbulence on those scales takes over, and the inertial-range turbulence enters the ion-inertia scales. If this happens, no reference is required to any sophisticated kind of hidden dissipation mechanism. Rather, this changing slope is quite a clear indication of the ion-inertial scale coming into play, clearer than the recalculation of scales via Taylor's hypothesis.

While the presented model is qualitatively similar to previous observations in that the magnetic energy spectra become steeper in the kinetic range, observed slopes are often steeper than $-7/3$, for example, as in Stawarz et al. (2016), Chen and Boldyrev (2017), and Breuillard et al. (2018). It should be noted that the theory predicts the energy spectra in the wave vector domain and the observations have often access to the spectra in the frequency domain. Possible reasons for the difference in the spectral slope between the theory and the observations include the presence of dispersive waves and the non-Gaussian frequency broadening in the random sweeping effect.

The turbulent Hall-electric power spectra directly map the turbulent velocity power spectra, the most important kinetic power spectra in any turbulence. Since these at short scales are very difficult to measure, the observation of Hall-turbulence should give a direct clue to their identification.

Hall turbulence quite strongly affects the inertial range turbulent density spectra on ion-inertial scales, as recently suggested (Treumann et al., 2019). Hall density power spectra increase in their most important compressive and thus dominant section as $k_\perp^{+5/3}$ which is an inverse Kolmogorov increase! They contribute to the earlier found deviation from inertial range slope. Observations should distinguish of its absolute contribution which from phenomenological scaling theory cannot be determined. The obtained steep spectral increase, when overlaid on ordinary spectra might contribute to the occasionally observed and still mysterious $k^{-1}$ spectral slopes.

The data analysis motivated model of Alexandrova et al. (2008) introduces an ad hoc measure $\alpha$ of the compression distinguishing between the incompressible ($\alpha = 0$) and isotropic compressible ($|\alpha| = 1$) cases. It maps the spectral slope of the magnetic field energy from $k^{-7/3}$ in the incompressible case to $(-7 + 6\alpha)/3$ in the compressible case. Our physically moti-

vated Hall MHD model differs from that of Alexandrova et al. (2008) in that the slope $-7/3$ is obtained for the compressible field fluctuations.

### Electrostatic nature

The Hall electric field attains the electrostatic component when the wave vectors are perpendicular or nearly perpendicular to the magnetic field. This applies to both the compressible and incompressible cases of magnetic fluctuations. The energy spectrum of the Hall electric field has a flatter spectral slope than that of the the magnetic field.

Care must be exercised when analysing electric field data and estimating the phase speed by reference to the E-B ratio, in particular, in the ion-kinetic range. In the compressible case the E-B ratio depends linearly on the wavenumber $k_\perp$ as considered earlier in Cluster data analysis (Matteini et al., 2017) and hybrid plasma simulations (Franci et al., 2015), while the incompressible case exhibits a $\sqrt{k_\perp}$ dependence, The character of the electric field needs to be evaluated when performing the E-B ratio analysis. It should be determined whether the electric field is of electromagnetic nature, representing a dispersive wave, or it is electrostatic, in which case it results from Hall-turbulence.

### Parallel vs. perpendicular components of the magnetic field

Some observations (Stawarz et al., 2016; Chen and Boldyrev, 2017) indicate the dominance of the perpendicular magnetic field component in the kinetic range. Our scaling laws are derived separately for the parallel one. It predicts that the Hall electric field associated with the parallel component of the magnetic field should dominate the electric spectrum (Eqs. 5–7). The magnetic energy spectrum can be dominated by either parallel or perpendicular fluctuations. However note that the scaling contains the undetermined numerical constant $c_m$ which determines the absolute value.

The parallel fluctuating component dominates if both compressive and incompressible fluctuations are excited by the electron flow. The normalised perpendicular component of the magnetic field is smaller than the parallel component according to $\delta\tilde{B}_\parallel \sim |\delta\tilde{B}_\perp|^2$. This follows from the electron flow velocity $\tilde{v}_\perp \sim \tilde{k}_\perp \delta\tilde{B}_\parallel$ and the association to the perpendicular component $\tilde{v}_\perp \sim \tilde{k}_\perp |\delta\tilde{B}_\perp|^2$. In Hall MHD the flow velocity is $\boldsymbol{E} \times \boldsymbol{B}$ passive being subject to the magnetic and electric fields.

The relative contribution between the parallel and perpendicular components of the magnetic field depends on the length scales. Using Eq. (21) and Eq. (33) yields

$$\frac{\delta B_\parallel}{\delta B_\perp} \propto \tilde{k}^{-1/6}. \tag{38}$$

Therefore, the contribution of the parallel component of the magnetic field is reduced with increasing wavenumber.

### Density spectrum

The increasing sense of the smaller-scale (or higher-frequency) density spectrum is indeed found using the Spektr-R spacecraft data in the solar wind (Šafránková et al., 2013). Treumann et al. (2019) provide a theoretical explanation of the density spectrum bump using the convected fluid model which the present theory extends to the inclusion of Hall dynamics. In the magnetosheath, to date no such increase is observed in the electron density spectrum based on the spacecraft data. Figure 3 in Breuillard et al.

(2018) shows a flattening of the density spectrum at spacecraft-frame frequencies of 10 Hz or higher, but this flattening is more likely associated with the Poisson noise in the particle measurements, indicating that clean, proper density spectrum measurements will be an important future task in the observational study of the Hall-domain physics. Chen and Boldyrev (2017, their Fig 4, bottom panel) shows that the density has about the same fluctuation power as the magnetic field at lower frequencies, indicating similar density and magnetic spectral slopes, with density spectrum estimated for electrons and inert ions. Theoretically information about the density spectrum can also be obtained either making use of the continuity equation or the quasi-static approximation (Cohen and Kulsrud, 1974; Narita and Hada, 2018).

**Gyro-kinetic treatment**

Schekochihin et al. (2009) provide a detailed description of ion-scale turbulence for weakly collisional plasmas through in a gyro-kinetic treatment. Gyro-kinetic theory is a reduced anisotropic limit of Hall-MHD with comparable results to that of the authors. However, the gyrokinetic theory, unlike Hall-MHD, incorporates phase-mixing due to Landau damping (not cyclotron-resonance). In weak turbulence of energy-cascading kinetic Alfvén waves gyro-kinetic theory predicts inertial-range energy spectra (in the perpendicular wavenumber domain) with spectral slopes $k_\perp^{-1/3}$ for the electric and $k_\perp^{-7/3}$ the magnetic fields, and spectral density slopes $k_\perp^{-7/3}$. These are identical to the compressive magnetic field fluctuations obtained here.

**Concluding remarks**

In summary, we believe that the detailed analysis of the particular properties of the Hall-inertial range turbulence contributes to the clarification of the behaviour of the plasma and electromagnetic field on the ion-inertia scales $k_\perp c/\omega_i > 1$, length scales shorter than that for the fluid or magnetohydrodynamic picture of turbulence. The wavenumber scaling laws and the corresponding power spectra are derived for Hall-turbulent magnetic, electric, velocity, and density field in the phenomenological approach. The Hall-inertial range is of great interest for many reasons in the both observational and theoretical sense.

In the observational studies of space plasma turbulence, various spectral observations have been performed in the past two decades and there is an increasing amount of evidence that the magnetic energy spectrum exhibits a dissipative sense (steeper sense) of the spectral curve. Occasionally it has even be named the dissipation (or ion-dissipation) range. Excitation of ion-kinetic electromagnetic waves (such as highly-oblique whistler mode, kinetic Alfvén mode, and ion Bernstein mode) is another possible scenario (which leads to the notion of dispersive range instead of dissipation range). Our model for the Hall-turbulence serves as a likely candidate to explain the steepening of the magnetic energy spectra neither as dissipation range nor as dispersive range but as Hall-inertial range.

In the theoretical studies, clarification of the spectral shapes in the Hall-inertial range should provide a useful background for the distinction among the inertial-range behaviour and dissipation of turbulence. Our Hall-turbulence model shows that the inertial-range can have a transition from fluid-scales (which is for MHD) to ion-scales (which is for Hall MHD) in a dissipation-less manner. The dissipation of turbulent fluctuations in collisionless plasmas remains poorly understood. The difference in the spectral shapes from Hall-inertial range would be interpreted as a sign of the onset of dissipation.

*Acknowledgements.* This work is financially supported by the Austrian Space Applications Programme (ASAP) at the Austrian Research Promotion Agency, FFG ASAP12 SOPHIE, under contract 853994.

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
