# Peer review of "Scaling laws in Hall-inertial range turbulence"

_Annales Geophysicae, 2019_

## Referee Comment (RC1) · Anonymous Referee #1 · 18 Jun 2019

**General comments**

The paper, "Scaling laws in Hall-inertial range turbulence", discusses the behavior of the Hall term from generalized Ohm's law in the context of turbulent systems. The authors focus on two-dimensional turbulence and develop phenomenological scaling laws for the wavenumber spectra of the magnetic field, electric field and particle number density between ion and electron scales. The authors find differences in the spectral behavior associated with the parallel (compressive) and perpendicular (incompressive) components of the magnetic fluctuations with respect to the mean magnetic field and qualitative similarities between the presented model and previous observations are noted. I find the paper to be well written and believe the analysis provides important insight into the role of the Hall term in turbulent plasmas, which may be useful for

interpreting future high-resolution measurements of space plasma turbulence. My main comment is that further discussion of the differences between the presented model and previous observations could be provided, which would give useful insight on the limitations of the presented model. Additionally, I have noted some statements which could use clarification and additional references that may be relevant.

**Specific Comments**

- Pg. 3, line 19 – 22: I am unclear how the compressible and incompressible components of the magnetic field are obtained from $\delta B_H \cdot \delta J_H$ and $\delta B_H \times (\delta E_H \times B)$. Additionally, how is it possible for the incompressible component of B to be anti-parallel to $\delta E_H$ if $\delta E_H$ is in the $k_\perp$ direction? Wouldn't this imply that the magnetic field has a divergence?

- Section 4: Some further discussion of differences between the presented theory and previous observations and the possible explanations for these differences may be useful.

  For example:

  -While the presented model is qualitatively similar to previous observations in that the magnetic energy spectra become steeper in the kinetic range, observed slopes are often steeper than -7/3 [for example as in Stawarz et al. JGR, 121, 11021, 2016; Chen et al. ApJ, 842, 122, 2017; Breuillard et al. ApJ, 859, 127, 2018].

  -Would the presented theory predict that the fluctuations in the parallel component of the magnetic field should dominate the spectrum? Some observations [for example Stawarz et al. JGR, 121, 11021, 2016 and Chen et al. ApJ, 842, 122, 2017] seem to show the perpendicular component dominating into the kinetic range.
[Figure]

-How does the predicted increasing density fluctuation spectra mesh with the observed decreasing density spectra in the kinetic range, for example as recently observed with MMS by Breuillard et al. [ApJ, 859, 127, 2018] and Chen et al. [ApJ, 842, 122, 2017]?

• Matteini et al. [MNRAS, 466, 945, 2017] and Franci et al. [ApJ, 812, 21, 2015] may also be relevant references to discuss, as they also consider the role of the Hall term in generating a linear ratio between the electric and magnetic field.

**Technical Corrections**

• Abstract and Section 2: While the 2D nature of the turbulence in the solar wind is mentioned in the introduction, I think it would be useful to explicitly state in the Abstract and in Section 2 that a 2D geometry (i.e. no parallel component of the wavevectors) is being considered in this paper.

• Pg. 1, line 21: For a 500 km/s flow speed, shouldn't the frequency range be 0.5 to 5 Hz?

• Pg. 4, line 5: I think the right-hand-side of eq. 5 should be positive instead of negative.

• Pg. 5, line 15: A citation for the author's previous publication which is being referred to would be helpful.

• Pg. 7, line 10: I think that "revoke" should be "invoke".

• Pg. 7, line 19-21: Can the authors provide a reference for the statement in the last sentence in Section 3.1.

---

## Referee Comment (RC2) · Anonymous Referee #2 · 19 Jun 2019

Narita et al. propose a phenomenological model of turbulence for ion kinetic scales on the basis of Hall-MHD. The paper is clearly written and coherent and appropriate for Annales Geophysicae. However, several relevant references are missing, including some that arrive to similar results using the same formalism (Hall-MHD). I would highly encourage the authors to write a more detailed discussion and introduction incorporating theoretical and observational results relavant to their work and to describe limitations of their model.

1.Contrary to what the authors are saying on L5, there are now strong indication of kinetic Alfven waves in the ion kinetic range, e.g. Roberts et al. GRL,45, 2018. The authors should include this reference.

2.Regarding the limits of Hall-MHD to describe ion kinetic scale turbulence, there is

plenty in the literature available. Hall MHD is valid in the limit where the electron temperature is much greater than the ions temperature and when the inverse of the linear transit time for an ion is much smaller than the turbulent frequency and the inverse of the linear transit time for an electron, respectively. Thus, in the instance where the temperature of the ion is finite, phase-mixing and damping of modes ought to be taken into account. Perhaps a good recent reference that can be added and discussed is that of Howes et al., Nonlinear Processes of Geophysics, 16, 2009.

3.Schekochihin et al., ApJ Supplement, (2007) provides a detailed description of ion-scale turbulence for weakly collisional plasmas through the use of gyro kinetic. Gyrokinetic is a reduced anisotropic limit of Hall-MHD with comparable results to that of the authors. However, Gyrokinetic, unlike Hall-MHD, incorporates phase-mixing due to Landau damping (not cyclotron-resonance). Can the authors incorporate in their discussion a comparison of their results with that of Schekochihin et al..

4. Chen et al., ApJ, 122, 2017, among many others, report magnetic energy spectra that are steeper for ion kinetic range. Can the authors incorporate a more detailed discussion incorporating observational evidence that are quantitatively different from their theory? Perhaps differences between Hall-MHD turbulent estimates and observations can be used to quantify the contributions of kinetic physics at the ion scale?

5. Alexandrova et al.[Small scale energy cascade of the solar wind turbulence] arrive to a similar scaling as that of the authors using Hall-MHD. Can the authors differentiate their work from that of Alexandrova et al.

---

## Author Comment (AC1) · 13 Jul 2019

Thank you very much for taking time for reading the manuscript and raising helpful comments. Reply to Referee-1 comments are as follows. We add the full version of the manuscript revision and the reply comments as a supplementary pdf file.

1. *General comments*

*The paper, "Scaling laws in Hall-inertial range turbulence", discusses the behavior of the Hall term from generalized Ohm's law in the context of turbulent systems. The authors focus on two-dimensional turbulence and develop phenomenological scaling laws for the wavenumber spectra of the magnetic field,*

[Figure]

*electric field and particle number density between ion and electron scales. The authors find differences in the spectral behavior associated with the parallel (compressive) and perpendicular (incompressive) components of the magnetic fluctuations with respect to the mean magnetic field and qualitative similarities between the presented model and previous observations are noted. I find the paper to be well written and believe the analysis provides important insight into the role of the Hall term in turbulent plasmas, which may be useful for interpreting future high-resolution measurements of space plasma turbulence. My main comment is that further discussion of the differences between the presented modeland previous observations could be provided, which would give useful insight on the limitations of the presented model. Additionally, I have noted some statements which could use clarification and additional references that may be relevant.*

**Reply**:

- Thank you very much for the positive evaluation of our work.

2. *Specific Comments*

*Pg. 3, line 19 − 22: I am unclear how the compressible and incompressible components of the magnetic field are obtained from $\delta B_H \cdot \delta J_H$ and $\delta B_H \times (\delta E_H \times B)$. Additionally, how is it possible for the incompressible component of B to be anti-parallel to $\delta E_H$ if $\delta E_H$ is in the $k_\perp$ direction? Wouldn't this imply that the magnetic field has a divergence?*

**Reply**:

- Right. Thank you for noticing this point. The fluctuating magnetic field is

projected by referring to the mean magnetic field, not to the Hall current. Otherwise the product j dot B makes the current helicity, which is beyond the scope of the manuscript. The Hall current $\delta \boldsymbol{j}_H$ is used when estimating the fluctuating electric field in our theory (in the lowest-order sense). The fluctuating magnetic field is expressed by the scaling law.

- We modify the first part of section 2.2 as follows (page 3, line 29 –page 4 line 10).

"The magnetic field fluctuations $\delta \boldsymbol{B}_H$ in Hall turbulence have two components, one compressive component $\delta B_\parallel$ parallel to the mean field $\boldsymbol{B}$ and the other perpendicular component $\delta B_\perp$. It is convenient to introduce an orthogonal coordinate system with base vectors $\boldsymbol{e}_1$ and $\boldsymbol{e}_2$ perpendicular to the mean field, and $\boldsymbol{e}_\parallel$ along the mean field. Moreover, we are free to choose the direction of the perpendicular wave vector, letting $\boldsymbol{e}_1$ refer to $\boldsymbol{k}_\perp$. The Hall magnetic field has no divergence, so it must be perpendicular to $\boldsymbol{k}$. This yields

$$\delta \boldsymbol{B} = (0, \delta B_\perp, \delta B_\parallel). \tag{3}$$

The fluctuation of the Hall electric field is given by

$$\delta \boldsymbol{E}_H = \frac{1}{en} \delta \boldsymbol{j}_H \times \boldsymbol{B} - \frac{\delta n}{n} \boldsymbol{E}. \tag{4}$$

The last term on the right containing the fluctuations in density and their contribution to $\delta \boldsymbol{E}_H$ is important only in the stationary frame where $\boldsymbol{E} \neq 0$. Using Ampère's law $\mu_0 \delta \boldsymbol{j} = \nabla \times \delta \boldsymbol{B}$ (from here on suppressing the index $H$ on the fluctuations when dealing exclusively with Hall fluctuations in Hall MHD) yields

$$\delta \boldsymbol{E} = \frac{1}{en\mu_0} \boldsymbol{B} \times (\nabla \times \delta \boldsymbol{B}). \tag{5}$$

"

3. *Section 4: Some further discussion of differences between the presented theory and previous observations and the possible explanations for these differences may be useful.*

   *For example:*

   *– While the presented model is qualitatively similar to previous observations in that the magnetic energy spectra become steeper in the kinetic range, observed slopes are often steeper than -7/3 [for example as in Stawarz et al. JGR, 121, 11021, 2016; Chen et al. ApJ, 842, 122, 2017; Breuillard et al. ApJ, 859, 127,2018].*

   **Reply**:

   - Agreed. We added the following text (page 11, line 17–22).

     "While the presented model is qualitatively similar to previous observations in that the magnetic energy spectra become steeper in the kinetic range, observed slopes are often steeper than $-7/3$, for example, as in Stawarz et al. (2016), Chen and Boldyrev (2017), and Breuillard et al. (2018). It should be noted that the theory predicts the energy spectra in the wave vector domain and the observations have often access to the spectra in the frequency domain. Possible reasons for the difference in the spectral slope between the theory and the observations include the presence of dispersive waves and the non-Gaussian frequency broadening in the random sweeping effect."

4. *– Would the presented theory predict that the fluctuations in the parallel component of the magnetic field should dominate the spectrum? Some observations*

*[for example Stawarz et al. JGR, 121, 11021, 2016 and Chen et al. ApJ, 842,122, 2017] seem to show the perpendicular component dominating into the kinetic range.*

**Reply**:

- No, not necessarily. The theory does not immediately predict that the parallel component of the magnetic field should dominate the energy spectrum because fluctuations in the parallel and perpendicular components of the magnetic field are modeled independently. Our theory predicts that the electric field (or the Hall effect, naively speaking) associated with the parallel component of the magnetic field should dominate the spectrum (Equations 5–7). The magnetic energy spectrum can be dominated either by the parallel fluctuations or by the perpendicular fluctuations. But a naive estimate indicates that the parallel fluctuating component may dominate if both compressive and incompressible fluctuations are excited by the electron flow.

- We added a subsection "Parallel vs. perpendicular components of the magnetic field" and discuss the competition between the parallel and perpendicular components of the magnetic field (page 12, line 13–26).

"Some observations (Stawarz et al., 2016; Chen and Boldyrev, 2017) indicate the dominance of the perpendicular magnetic field component in the kinetic range. Our scaling laws are derived separately for the parallel one. It predicts that the Hall electric field associated with the parallel component of the magnetic field should dominate the electric spectrum (Eqs. 5–7). The magnetic energy spectrum can be dominated by either parallel or perpendicular fluctuations. However note that the scaling contains the undetermined numerical constant $c_m$ which determines the absolute value.

The parallel fluctuating component dominates if both compressive and incompressible fluctuations are excited by the electron flow. The normalised perpendicular component of the magnetic field is smaller than the parallel component according to $\delta\tilde{B}_\parallel \sim |\delta\tilde{B}_\perp|^2$. This follows from the electron flow velocity $\tilde{v}_\perp \sim \tilde{k}_\perp \delta\tilde{B}_\parallel$ and the association to the perpendicular component $\tilde{v}_\perp \sim \tilde{k}_\perp |\delta\tilde{B}_\perp|^2$. In Hall MHD the flow velocity is $\boldsymbol{E} \times \boldsymbol{B}$ passive being subject to the magnetic and electric fields.

The relative contribution between the parallel and perpendicular components of the magnetic field depends on the length scales. Using Eq. (21) and Eq. (33) yields

$$\frac{\delta B_\parallel}{\delta B_\perp} \propto \tilde{k}^{-1/6}. \qquad (6)$$

Therefore, the contribution of the parallel component of the magnetic field is reduced with increasing wavenumber."

5. – *How does the predicted increasing density fluctuation spectra mesh with the observed decreasing density spectra in the kinetic range, for example as recently observed with MMS by Breuillard et al. [ApJ, 859, 127, 2018] and Chen et al.[ApJ, 842, 122, 2017]?*

**Reply**:

- We added the following text (page 13, line 6–12).

  "The increasing sense of the smaller-scale (or higher-frequency) density spectrum is indeed found using the Spektr-R spacecraft data in the

solar wind (Šafránková et al., 2013). Treumann et al. (2019) provide a theoretical explanation of the density spectrum bump using the convected fluid model which the present theory extends the the inclusion of Hall dynamics. In the magnetosheath, to date no such increase is observed in the electron density spectrum based on the spacecraft data. Figure 3 in Breuillard et al. (2018) shows a flattening of the density spectrum at spacecraft-frame frequencies of 10 Hz or higher. Chen and Boldyrev (2017, Fig 4, bottom panel) shows that the density has about the same fluctuation power as the magnetic field at lower frequencies, indicating similar density and magnetic spectral slopes, with density spectrum estimated for electrons and inert ions. Theoretically information about the density spectrum can also be obtained either making use of the continuity equation or the quasi-static approximation (Cohen and Kulsrud, 1974; Narita and Hada, 2018)."

6. *Matteini et al. [MNRAS, 466, 945, 2017] and Franci et al. [ApJ, 812, 21, 2015] may also be relevant references to discuss, as they also consider the role of the Hall term in generating a linear ratio between the electric and magnetic field.*

**Reply**:

- Agreed. We added the following text (page 12, line 8–9).

  "...as considered earlier in Cluster data analysis (Matteini et al., 2017) and hybrid plasma simulation (Franci et al., 2015)"

7. *Technical Corrections*

*Abstract and Section 2: While the 2D nature of the turbulence in the solar wind is mentioned in the introduction, I think it would be useful to explicitly state in the Abstract and in Section 2 that a 2D geometry (i.e. no parallel component of the wavevectors) is being considered in this paper.*

**Reply**:

- Done. We added the sentence in abstract and section 1 (not section 2). (page 1, line 6–7; page 2, line 24–26).

  "In the present paper we consider a two-dimensional geometry with no wave vector component parallel to the magnetic field as is appropriate in Hall-MHD."

  "We consider a two-dimensional geometry which has no parallel wave vector component. The full expression for the Hall electric field contains also parallel wave vector components Treumann et al. (2019) which in Hall MHD are neglected."

8. *Pg. 1, line 21: For a 500 km/s flow speed, shouldn't the frequency range be 0.5to 5 Hz?*

**Reply**:

- Done. (page 1, line 22).

9. *Pg. 4, line 5: I think the right-hand-side of eq. 5 should be positive instead ofnegative.*

**Reply**:

- Right! Thank you! Corrected. (page 3, equations 3–11).

10. *Pg. 5, line 15: A citation for the author's previous publication which is being referred to would be helpful.*

    **Reply**:

    Done, "Treumamnn et al., 2019)" (page 5, line 26).

11. *Pg. 7, line 10: I think that "revoke" should be "invoke".*

    **Reply**:

    - Done. (page 7, line 18).

12. *Pg. 7, line 19–21: Can the authors provide a reference for the statement in the last sentence in Section 3.1.*

    **Reply**:

    - We added a reference to Šafránková et al. (ApJ, 2015). (page 8, line 2).

Please also note the supplement to this comment:
https://www.ann-geophys-discuss.net/angeo-2019-69/angeo-2019-69-AC1-supplement.pdf

[Figure]

**Supplement:**

[revised manuscript text omitted]

---

## Author Comment (AC2) · 13 Jul 2019

Thank you very much for taking time for reading the manuscript and raising helpful comments. Reply to Referee-2 comments are as follows. Please refer to the full version of the manuscript revision with the reply comments uploaded as a supplementary pdf file to the reply to the Referee-1 report.

1. *Narita et al. propose a phenomenological model of turbulence for ion kinetic scaleson the basis of Hall-MHD. The paper is clearly written and coherent and appropriate for Annales Geophysicae. However, several relevant references are missing, including some that arrive to similar results using the same formalism (Hall-MHD). I would highly encourage the authors to write a more detailed*

[Figure]

*discussion and introduction incorporating theoretical and observational results relavant to their work and to describe limitations of their model.*

*1. Contrary to what the authors are saying on L5, there are now strong indication of kinetic Alfven waves in the ion kinetic range, e.g. Roberts et al. GRL, 45, 2018. The authors should include this reference.*

**Reply**:

- We agree that Roberts et al. (GRL, 2018) indicate the kinetic Alfvén mode in the magnetosheath region from the wave analysis using the fluctuation sense such as the Alfven ratio or the ratio of the ion density to the magnetic field, not from the dispersion relation analysis. Narita et al. (2016) shows, on the other hand, that there is a freuqency scattering in the observationally-determined dispersion relation with an indication to kinetic-drift mirror mode.

- We added the following text. (page 2, line 16–19).
  "The study by Roberts et al. (2018) indicates the existence of the kinetic Alfvén mode in the magnetosheath region from the wave analysis for the fluctuations in the MMS data such as the Alfven ratio. No dispersion analysis is performed. On the other hand, the study by Narita et al. (2016) exhibits a frequency scattering in the observationally-determined dispersion relation with an indication of a kinetic-drift mirror mode."

2. *Regarding the limits of Hall-MHD to describe ion kinetic scale turbulence, there is plenty in the literature available. Hall MHD is valid in the limit where the electron temperature is much greater than the ions temperature and when the inverse of the linear transit time for an ion is much smaller than the turbulent frequency and the inverse ofthe linear transit time for an electron, respectively. Thus, in the*

*instance where thetemperature of the ion is finite, phase-mixing and damping of modes ought to be taken into account. Perhaps a good recent reference that can be added and discussed is that of Howes et al., Nonlinear Processes of Geophysics, 16, 2009.*

**Reply**:

- Agreed. We added the following text on page 2, line 27–31.

  "Limitations of Hall MHD have been discussed, for example, by Howes (2009). The concept of Hall turbulence is valid in the limit where the electron temperature is much greater than the ions temperature and when the inverse of the linear transit time for an ion is much smaller than the turbulent frequency and the inverse of the linear transit time for an electron, respectively. Thus, in the instance where the temperature of the ions is finite, phase-mixing and damping of modes ought to be taken into account. This causes deviations from Hall MHD."

3. *Schekochihin et al., ApJ Supplement, (2007) provides a detailed description of ion-scale turbulence for weakly collisional plasmas through the use of gyro kinetic. Gyrokinetic is a reduced anisotropic limit of Hall-MHD with comparable results to that of the authors. However, Gyrokinetic, unlike Hall-MHD, incorporates phase-mixing due to Landau damping (not cyclotron-resonance). Can the authors incorporate in the discussion of a comparison of their results with that of Schekochihin et al..*

**Reply**:

- Agreed. Done. Schekochihin et al. (2007) should read Schekochihin et al.

(2009). We added the following text (page 13, line 6–12).

"Schekochihin et al. (2009) provide a detailed description of ion-scale turbulence for weakly collisional plasmas through in a gyro-kinetic treatment. Gyro-kinetic theory is a reduced anisotropic limit of Hall-MHD with comparable results to that of the authors. However, the gyrokinetic theory, unlike Hall-MHD, incorporates phase-mixing due to Landau damping (not cyclotron-resonance). In weak turbulence of energy-cascading kinetic Alfvén waves gyro-kinetic theory predicts inertial-range energy spectra (in the perpendicular wavenumber domain) with spectral slopes $k_\perp^{-1/3}$ for the electric and $k_\perp^{-7/3}$ the magnetic fields, and spectral density slopes $k_\perp^{-7/3}$. These are identical to the compressive magnetic field fluctuations obtained here."

4. *Chen et al., ApJ, 122, 2017, among many others, report magnetic energy spectra that are steeper for ion kinetic range. Can the authors incorporate a more detailed discussion incorporating observational evidence that are quantitatively different from their theory? Perhaps differences between Hall-MHD turbulent estimates and observationscan be used to quantify the contributions of kinetic physics at the ion scale?*

**Reply**:

- The same question was raised by Referee-1. We added the following text (page 11, line 17–22).

[revised manuscript text omitted]

---

## Author Response (AR2)

**Reply to referee comments**

Manuscript ID: angeo-2019-69
Scaling laws inHall-inertial range turbulence
Y. Narita et al.
* * *
**Referee 1**

1. *I find that the authors have reasonably addressed my previous comments and that the manuscript has improved as a result of the revisions. However, I do have several minor comments which I think should be addressed.*

   *Line 11 of page 1: I wonder if the statement that the model presented in this study is the likely explanation for the steepening of the magnetic spectrum is a bit strong, given that there are several discrepancies pointed out (for example different power laws, different behavior of the density, etc.). Perhaps noting that the Hall inertial range is a possible explanation would be better.*

   *Line 30 of page 12: "extends the the inclusion" should be "extends to the inclusion"*

   *Line 1 of page 13: The authors state that a flattening of the density spectrum is present in Figure 3 of Breuillard et al. 2018 above 10 Hz, implying that this may relate to the expected increase due to hall physics. However, I believe that this flattening is more likely associated with the Poisson noise in the FPI measurements, as was also noted by Breuillard et al.*

   **Reply**:

   - Thank you very much for taking time to carefully check the revised manuscript. We gladly work on the minor comments and deliver herewith the manuscript in a hope of being forwarded to production.

   - Line 11 of page 1. Agreed. We change the sentence "Our model..." (in the abstract field) as follows (page 1, line 10–12 in the second revision).

     "Our model for the Hall-turbulence gives a possible explanation for the steepening of the magnetic energy spectra in the solar wind neither as indication of the dissipation range nor the dispersive range but as the Hall-inertial range."

   - Line 30 of page 12. Done (page 12, line 30 in the second revision).

   - Line 1 of page 13. Agreed. We change the sentence "Figure 3..." as follow (page 12, line 31 to page 13, line 3 in the second revision).

     "Figure 3 in Breuillard et al. (2018) shows a flattening of the density

spectrum at spacecraft-frame frequencies of 10 Hz or higher, but this flattening is more likely associated with the Poisson noise in the particle measurements, indicating that clean, proper density spectrum measurements will be an important future task in the observational study of the Hall-domain physics."
* * *
**Referee 2**

1. *accepted as is.*

   **Reply**:

   - Thank you.